# Expander Graph Propagation

**Andreea Deac**
Mila, Université de Montréal
deacandr@mila.quebec

**Marc Lackenby**
University of Oxford
lackenby@maths.ox.ac.uk

**Petar Veličković**
DeepMind
petarv@deepmind.com

## Abstract

Deploying graph neural networks (GNNs) on whole-graph classification or regression tasks is known to be challenging: it often requires computing node features that are mindful of both local interactions in their neighbourhood and the global context of the graph structure. GNN architectures that navigate this space need to avoid pathological behaviours, such as bottlenecks and oversquashing, while ideally having linear time and space complexity requirements. In this work, we propose an elegant approach based on propagating information over *expander graphs*. We leverage an efficient method for constructing expander graphs of a given size, and use this insight to propose the EGP model. We show that EGP is able to address all of the above concerns, while requiring minimal effort to set up, and provide evidence of its empirical utility on relevant graph classification datasets and baselines in the Open Graph Benchmark. Importantly, using expander graphs as a template for message passing necessarily gives rise to negative curvature. While this appears to be counterintuitive in light of recent related work on oversquashing, we theoretically demonstrate that negatively curved edges are likely to be **required** to obtain scalable message passing without bottlenecks. To the best of our knowledge, this is a previously unstudied result in the context of graph representation learning, and we believe our analysis paves the way to a novel class of scalable methods to counter oversquashing in GNNs.

## 1 Introduction

Graph neural networks (GNNs) are a flexible class of models for learning representations over graph-structured data [1]. Their versatility [2–4] and generality [5, 6] has made them a very attractive approach, leading to considerable application in areas as diverse as virtual drug screening [7], traffic prediction [8], combinatorial chip design [9] and pure mathematics [10, 11].

Most GNNs rely on repeatedly propagating information between neighbouring nodes in the graph. This is commonly expressed in the *message passing* [4] paradigm: nodes send vector-based *messages* to each other along the edges of the graph, and nodes update their representations by *aggregating* all the messages sent to them, in a permutation-invariant manner. Under many industrially-relevant tasks, this paradigm is very potent, often allowing for highly scalable model variants [12–14].

However, in many areas of scientific interest, purely local interactions are likely insufficient. Among the principal graph tasks, *graph classification* is perhaps most ripe with such situations: to meaningfully attach a label to a graph, in many cases it is insufficient to treat graphs as "bags of nodes". For example, when classifying a molecule for its potency as a candidate drug [7], the label is driven by complex substructure interactions in the molecule [15], rather than a naïve sum of atom-level effects.

Accordingly, GNNs deployed in this regime need to update node features in a manner that is mindful of the *global* properties of the graph. It quickly became apparent that it is often inadequate to merely stack more message passing layers over the input graph. In fact, for many graph classification tasks, such approaches may be weaker than discarding the graph structure altogether [16, 17]. Now, it is well-understood that stacking many local layers leaves GNNs vulnerable to pathological behaviours such as oversquashing [18]. Intuitively, oversquashing occurs when nodes need to store quantities of

A. Deac, M. Lackenby and P. Veličković, Expander Graph Propagation. *Proceedings of the First Learning on Graphs Conference (LoG 2022)*, PMLR 198, Virtual Event, December 9–12, 2022.

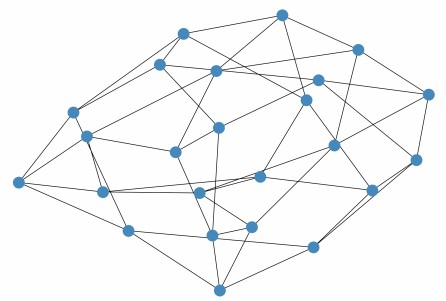 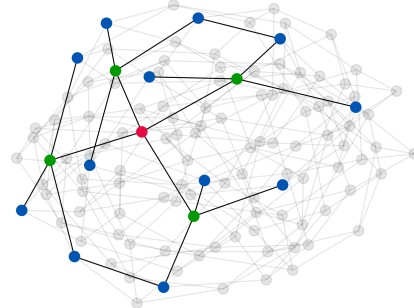

**Figure 1: Left**: The Cayley graph of $\mathrm{SL}(2,\mathbb{Z}_3)$, constructed using our method. It has $|V| = 24$ nodes and it is $4$-regular (implying $|E| = 2|V|$), hence it is sparse. Despite its sparsity, it is highly interconnected: any node is reachable from any other node by no more than $4$ hops. Hence, it can serve as a strong "template" for globally propagating node features with a GNN. **Right**: The Cayley graph of $\mathrm{SL}(2,\mathbb{Z}_5)$, constructed in an analogous way (with $|V| = 120$ nodes). A 2-hop neighbourhood of one node (in red) is highlighted, demonstrating its tree-like local structure.

information that are *exponentially* increasing with model depth [18, Section 5]. Such nodes often arise in the vicinity of *bottlenecks* in a graph—small collections of edges which are responsible for carrying representations between large groups of nodes. One typical example of such a bottleneck can be found in a *barbell graph* 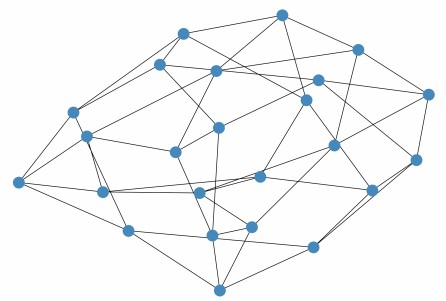, where the red edge is under significant representational pressure to transport information between the two communities.

Within this space, we are interested in proposing a method that satisfies *four* desirable criteria: (**C1**) it is capable of propagating information *globally* in the graph; (**C2**) it is *resistant* to the oversquashing effect and does not introduce bottlenecks; (**C3**) its time and space complexity remain *subquadratic* (tighter than $O(|V|^2)$ for sparse graphs); and (**C4**) it requires *no dedicated preprocessing* of the input. Satisfying all four of these criteria simultaneously is challenging, and we will survey many of the popular approaches in the next section—demonstrating ways in which they fail to meet some of them.

In this paper, we identify *expander graphs* as very attractive objects in this regard. Specifically, they offer a family of graph structures that are fundamentally *sparse* ($|E| = O(|V|)$), while having *low diameter*: thus, any two nodes in an expander graph may reach each other in a short number of hops, eliminating bottlenecks and oversquashing (see Figure 1). Further, we will demonstrate an efficient way to construct a family of expander graphs (leveraging known theoretical results on the *special linear group*, $\mathrm{SL}(2,\mathbb{Z}_n)$). Once an expander graph of appropriate size is constructed, we can perform a certain number of GNN *propagation* steps over its structure to globally distribute the nodes' features. Accordingly, we name our method *expander graph propagation* (**EGP**).

A key contribution of our work extends the implications of prior art on oversquashing via curvature analysis [19]. According to [19], negatively curved edges are causing the oversquashing effect—yet, counterintuitively, the edges of the expander graphs we construct will *always be negatively curved*! We prove, however, that our expanders can never be sufficiently negatively curved to trigger the conditions necessary for the results in [19] to be applicable, and show that the existence of negatively curved edges might in fact be **required** in order to have sparse communication without bottlenecks.

## 2   Related work

We begin with a survey of the many prior approaches to handling global context in graph representation learning, evaluating them carefully against our four desirable criteria (**C1–C4**; cf. Table 1). This list is by no means exhaustive, but should be indicative of the most important directions.

**Stacking more layers.** As already highlighted, one way to achieve global information propagation is to have a deeper GNN. In this case, we are capable of satisfying (**C1**) and (**C4**)—no dedicated preprocessing is needed. However, depending on the graph's diameter, we may need up to $O(|V|)$ layers to cover the graph, leading to quadratic complexity (violating (**C3**)) and introducing a vulnerability to bottlenecks (**C2**), as theoretically and empirically demonstrated in [18].

**Table 1:** A summary of principal approaches to handling global context in graph representation learning (Section 2). "(✓)" indicates that a criterion *may* be satisfied, depending on the method's tradeoffs. Our proposal, the expander graph propagation (EGP) method, satisfies all four criteria.

| Approach | (C1) (global prop.) | (C2) (no bottlenecks) | (C3) (subquadratic) | (C4) (no dedicated preproc.) |
|---|:---:|:---:|:---:|:---:|
| GNNs | ✗ | ✗ | ✓ | ✓ |
| Sufficiently deep GNNs | ✓ | ✗ | ✗ | ✓ |
| Master node [4, 20] | ✓ | ✗ | ✓ | ✓ |
| Fully connected [18, 21–25] | ✓ | ✓ | ✗ | ✓ |
| Feature aug. [26–31] | ✓ | (✓) | (✓) | ✗ |
| Graph rewiring [19, 32, 33] | ✓ | ✓ | ✓ | ✗ |
| Hierarchical MP [34–39] | ✓ | ✓ | (✓) | ✗ |
| **EGP** (ours) | ✓ | ✓ | ✓ | ✓ |

**Master nodes.** An attractive approach to introducing global context is to introduce a *master node* to the graph, and connect it to all of the graph's nodes. This can be done either explicitly [4] or implicitly, by storing a "global" vector [20]. It trivially reduces the graph's diameter to 2, introduces $O(1)$ new nodes and $O(|V|)$ new edges, and requires no dedicated preprocessing, hence it satisfies (**C1**, **C3**, **C4**). However, these benefits come at the expense of introducing a bottleneck in the master node: it has a very challenging task (especially when graphs get larger) to continually incorporate information over a very large neighbourhood in a useful way. Hence it fails to satisfy (**C2**).

**Fully connected graphs.** The converse approach is to make *every* node a master node: in this case, we make all pairs of nodes connected by an edge—this was initially proposed as a powerful method to alleviate oversquashing by [18]. This strategy proved highly popular in the recent surge of Graph Transformers [22, 23, 25], and is common for GNNs used in physical simulation [21] or reasoning [24] tasks. The graph's diameter is reduced to 1, no bottlenecks remain, and the approach does not require any dedicated preprocessing. Hence (**C1**, **C2**, **C4**) are trivially satisfied. The main downside of this approach is the introduction of $O(|V|^2)$ edges, which means (**C3**) can never be satisfied—and this approach will hence be prohibitive even for modestly-sized graphs.

**Feature augmentation.** An alternative approach is to provide additional features to the GNN which directly identify the structural role each node plays in the graph [26]. If done properly (i.e., if the computed features are relevant to the target), this can drastically improve expressive power. Hence, in theory, it is possible to satisfy (**C1**) while not violating (**C2**, **C3**). However, computing appropriate features requires either specific domain knowledge, or appropriate pre-training [27–31], in order to obtain such embeddings. Hence all of these gains come at the expense of failing to satisfy (**C4**).

**Graph rewiring.** Another promising line of research involves modifying the edges of the original graph to alleviate bottlenecks. Popular examples of this approach involve using diffusion [32]—which diffuse additional edges through the application of kernels such as the personalised PageRank, and stochastic discrete Ricci flows [19]—which surgically modify a small quantity of edges to alleviate the oversquashing effect on the nodes with negative Ricci curvature. Recent concurrent work [33] also uses constructions inspired by expander graphs to randomly locally rewire a given input graph. If realised carefully, such approaches will not deviate too far from the original graph, while provably alleviating oversquashing; hence it is possible to satisfy (**C1**, **C2**, **C3**). However, this comes at a cost of having to examine the input graph structure, with methods that do not necessarily scale easily with the number of nodes. As such, dedicated preprocessing is needed, failing to satisfy (**C4**).

**Hierarchical message passing.** Lastly, going beyond modifying the edges, it is also possible to introduce additional *nodes* in the graph—each of them responsible for a particular *substructure* in the graph[1]. If done carefully, it has the potential to drastically reduce the graph's diameter while not introducing bottlenecked nodes (hence, allowing us to satisfy (**C1**, **C2**)). However, in prior work, a cost has to be paid for this, usually in the need for dedicated preprocessing. Prior proposals for hierarchical GNNs that remain scalable require a dedicated pre-processing step [34–36], sometimes coupled with domain knowledge [36]—thus failing to satisfy (**C4**). In addition, such methods may

---

[1]Master nodes are a special case: a single node is responsible for a "substructure" spanning the entire graph.

require adding prohibitively large numbers of substructures [37, 38] or expensive pre-computation, e.g. computing the graph Laplacian eigenvectors [39]. This might make even (**C3**) hard to satisfy.

We remark that our work is not the first to study expander graph-related topics in the context of GNNs. Specifically, the ExpanderGNN [40] leverages expander graphs over neural network weights to sparsify the update step in GNNs. This is a direct application of Deep Expander Networks [41], which studied such constructs over CNNs. With respect to our contributions, neither of these cases discuss expanders in the context of the computational graph for a GNN, nor attempt to propagate messages over such a structure. Further, neither satisfy all four of our desired criteria (**C1**–**C4**).

## 3 Theoretical background

We now dedicate our attention to the key theoretical results over expander graphs, which will allow EGP to have favourable properties and be efficiently precomputable.

**Definition 1.** For a finite connected graph $G = (V(G), E(G))$, we consider functions $f \colon V(G) \to \mathbb{R}$. The *Laplacian* $Lf \colon V(G) \to \mathbb{R}$ of such a function is defined to be

$$Lf(v) = \deg(v)f(v) - \sum_{vw \in E(G)} f(w),$$

where $\deg(v)$ is the degree of the vertex $v$.

The mapping $L \colon \mathbb{R}^{V(G)} \to \mathbb{R}^{V(G)}$ sending a function $f$ to its Laplacian $Lf$ is a linear transformation. It is not hard to show [42] that $L$ is symmetric with respect to the standard basis for $\mathbb{R}^{V(G)}$ and positive semi-definite and hence has non-negative real eigenvalues

$$0 = \lambda_0(G) < \lambda_1(G) \leqslant \lambda_2(G) \leqslant \dots.$$

The smallest eigenvalue is $0$ and its associated eigenspace consists of the constant functions (assuming $G$ is connected). The smallest positive eigenvalue, $\lambda_1(G)$, is central to the definition of expander graphs, as the next definition shows.

**Definition 2.** An infinite collection $\{G_i\}$ of finite connected graphs is an *expander family* if there is a constant $c > 0$ such that for all $G_i$ in the collection, $\lambda_1(G_i) \geqslant c$.

Expander families [43–45] have many remarkable and useful properties, particularly when there is a uniform upper bound on the degree of the vertices of $G_i$.

**Definition 3.** Let $G$ be a finite graph. For $A \subset V(G)$, its *boundary* $\partial A$ is the collection of edges with one endpoint in $A$ and one endpoint not in $A$. The *Cheeger constant* $h(G)$ is defined to be

$$h(G) = \min \left\{ \frac{|\partial A|}{|A|} : A \subset V(G), 0 < |A| \leqslant |V(G)|/2 \right\}.$$

Thus, having a small Cheeger constant is equivalent to the graph having a 'bottleneck', in the sense that there is a collection of edges $\partial A$ that, when removed, disconnects the vertices into two sets ($A$ and its complement, $V(G) \backslash A$), with the property that the sizes of $A$ and its complement are significantly larger than the size of $\partial A$.

Expander families can be reinterpreted using Cheeger constants, as follows (see, e.g., [46–49]):

**Theorem 4.** *Let $\{G_i\}$ be an infinite collection of finite connected graphs with a uniform upper bound on their vertex degrees. Then the following are equivalent:*

1. *$\{G_i\}$ is an expander family;*

2. *there is a constant $\epsilon > 0$ such that for all graphs in the collection, $h(G_i) \geqslant \epsilon$.*

Hence, expander graphs have higher Cheeger constants and will hence experience less severe problems arising due to bottleneck edges. The following result is one of the many useful properties of expander families, and it concerns their *diameter*. It was proved by Mohar [50, Theorem 2.3]. See also [47].

**Theorem 5.** *The diameter* $\mathrm{diam}(G)$ *of a graph $G$ satisfies*

$$\mathrm{diam}(G) \leqslant 2 \left\lceil \frac{\Delta(G) + \lambda_1(G)}{4\lambda_1(G)} \log(|V(G)| - 1) \right\rceil,$$

*where $\Delta(G)$ is the maximal degree of any vertex of G. Hence, if $\{G_i\}$ is an expander family of finite graphs with a uniform upper bound on their vertex degrees, then there is a constant $k > 0$ such that for all graphs in the family,*

$$\mathrm{diam}(G_i) \leqslant k \log V(G_i).$$

Therefore, if we want to globally propagate information over an expander graph which has $|V|$ nodes, we only need $O(\log |V|)$ propagation steps to do so—yielding subquadratic complexity.

We showed that expanders will experience less severe problems arising due to bottleneck edges, with favourable propagation qualities. What is missing is an efficient method of constructing an expander of (roughly) $|V|$ nodes. To demonstrate such a method, we leverage known results from group theory.

**Definition 6.** A group $(\Gamma, \circ)$ is a set $\Gamma$ equipped with a *composition* operation $\circ : \Gamma \times \Gamma \to \Gamma$ (written concisely by omitting $\circ$, i.e. $g \circ h = gh$, for $g, h \in \Gamma$), satisfying the following axioms:

- *(Associativity)* $(gh)l = g(hl)$, for $g, h, l \in \Gamma$.

- *(Identity)* There exists a unique $e \in \Gamma$ satisfying $eg = ge = g$ for all $g \in \Gamma$.

- *(Inverse)* For every $g \in \Gamma$ there exists a unique $g^{-1} \in \Gamma$ such that $gg^{-1} = g^{-1}g = e$.

A group is hence a natural construct for reasoning about transformations that leave an object invariant (unchanged). Further, we define a relevant notion of a group's generating set:

**Definition 7.** Let $\Gamma$ be a group. A subset $S \subseteq \Gamma$ is a *generating set* for $\Gamma$ if it can be used to "generate" all of $\Gamma$ via composition. Concretely, any element $g \in \Gamma$ can be expressed by composing elements in the generating set, or their inverses; that is, we can express $g = s_1^{\pm 1} s_2^{\pm 1} s_3^{\pm 1} \cdots s_{n-1}^{\pm 1} s_n^{\pm 1}$ for $s_i \in S$.

Now we are ready to define a Cayley graph of a group w.r.t. its generating set.

**Definition 8.** Let $\Gamma$ be a group with a finite generating set $S$. Then the associated *Cayley graph* $\mathrm{Cay}(\Gamma; S)$ has vertex set $\Gamma$ and it has an edge $g \to gs$ for each $g \in \Gamma$ and each $s \in S$. We say that $s$ is the *label* on this edge. This is a potentially non-simple graph, as it may have edges with both endpoints on the same vertex and it may have multiple edges between a pair of vertices. In particular, when $s$ has order 2, then we view the edge $g \to gs$ and the edge $gs \to gs^2 = g$ as distinct edges.

Note that the degree of each vertex of a Cayley graph $\mathrm{Cay}(\Gamma; S)$ is $2|S|$. This is because each vertex $g$ is joined by edges to $gs$ and $gs^{-1}$ for each $s \in S$. Thus, we shall be particularly interested in the case where there is a uniform upper bound on $|S|$. The specific group we use for EGP is as follows.

For each positive integer $n$, the *special linear group* $\mathrm{SL}(2, \mathbb{Z}_n)$ denotes the group of $2 \times 2$ matrices with entries that are integers modulo $n$ and with determinant 1. One of its generating sets is:

$$S_n = \left\{ \begin{pmatrix} 1 & 1 \\ 0 & 1 \end{pmatrix}, \begin{pmatrix} 1 & 0 \\ 1 & 1 \end{pmatrix} \right\}.$$

Central to our constructions is the following important result.

**Theorem 9.** *The family of Cayley graph* $\mathrm{Cay}(\mathrm{SL}(2, \mathbb{Z}_n); S_n)$ *forms an expander family.*

The proof uses a result of Selberg [51] who showed that the smallest positive eigenvalue of the Laplacian of certain hyperbolic surfaces is at least $3/16$. One can use this to a produce a lower bound on the first eigenvalue of the Laplacian on $\mathrm{Cay}(\mathrm{SL}(2, \mathbb{Z}_n); S_n)$. Full proofs are given in [44, 45].

Lastly, it is useful to state a known result: the number of nodes of $\mathrm{Cay}(\mathrm{SL}(2, \mathbb{Z}_n); S_n)$ is:

$$|V(\mathrm{Cay}(\mathrm{SL}(2, \mathbb{Z}_n); S_n))| = n^3 \prod_{\text{prime } p|n} \left( 1 - \frac{1}{p^2} \right), \tag{10}$$

hence, it is of the order of $O(n^3)$. We now study the local properties of Cayley graphs in detail.

## 4   Local structure of the Cayley graphs, and the utility of negative curvature

Recent work [19] has suggested that the local structure of the graph $G$ underlying a GNN may play an important role in the way that information propagates around $G$. In particular, various notions of 'Ricci curvature' such as Forman curvature [52], Ollivier curvature [53, 54] and balanced Forman

curvature [19] have been examined. These are all local quantities, in the sense that they depend on the structure of the graph within a small neighbourhood of each edge. In this section, we will therefore examine the local structure of the Cayley graphs $G_n = \mathrm{Cay}(\mathrm{SL}(2, \mathbb{Z}_n); S_n)$.

The various notions of curvature given above are defined for each $e$ of the graph $G$. Since, as defined by [19], the balanced Forman curvature of an edge depends only on local structures (i.e. triangles and squares) around that edge, they can be determined by only observing the immediate 2-hop surrounding of that edge. Formally, for an edge $e$ of a graph $G$, let $N_2(e)$ be the induced subgraph with vertices that are at most two hops away from at least one endpoint of $e$. Then the curvature of $e$ only depends on the isomorphism type of $N_2(e)$. More specifically, if $e$ and $e'$ are edges in possibly distinct graphs, and there is a graph isomorphism between $N_2(e)$ and $N_2(e')$ that sends $e$ to $e'$, then this guarantees that the curvatures of $e$ and $e'$ are equal.

This situation arises prominently in the Cayley graphs that we are considering, as follows.

**Proposition 11.** *Let $s$ be one of*

$$\begin{pmatrix} 1 & 1 \\ 0 & 1 \end{pmatrix}, \qquad \begin{pmatrix} 1 & 0 \\ 1 & 1 \end{pmatrix}.$$

*Let $n, n' > 18$ and let $e$ and $e'$ be $s$-labelled edges in $G_n$ and $G_{n'}$. Then there is a graph isomorphism between $N_2(e)$ and $N_2(e')$ taking $e$ to $e'$.*

We prove Proposition 11 in Appendix A. This immediately allows us to characterise the balanced Forman curvature and Ollivier curvature for all of the Cayley graphs we generate:

**Proposition 12.** *The balanced Forman curvatures* $\mathrm{Ric}(n)$*, and the Ollivier curvatures $\kappa(n)$ of all edges of Cayley graphs $G_n$ are given by:*

$$\mathrm{Ric}(n) = \begin{cases} 0 & \text{if } n = 2 \\ -1/4 & \text{if } n = 3 \\ -1/2 & \text{if } n = 4 \\ -1 & \text{if } n \geqslant 5, \end{cases} \qquad \kappa(n) = \begin{cases} 0 & \text{if } n = 2 \\ -1/8 & \text{if } n = 3 \\ -1/4 & \text{if } n = 4 \\ -3/8 & \text{if } n = 5 \\ -1/2 & \text{if } n \geqslant 6. \end{cases}$$

*Proof.* Proposition 11 implies that the balanced Forman and Ollivier curvatures are all equal for $n > 18$. Their values for $2 \leqslant n \leqslant 19$ can all be empirically computed, and are given as above. $\quad\square$

Prior work [19] suggests it is preferable for GNNs to operate on graphs with positive Ricci curvature, whereas our graphs $G_n$ $(n > 2)$ all have negative Ricci curvature. However, we contend that negative Ricci curvature is not in itself an impediment to efficient propagation around a GNN. Indeed, it was shown in [19, Theorem 4] that poor propagation arises when the balanced Forman curvature is close to $-2$, specifically if it is at most $-2 + \delta$ for some $\delta > 0$. Here, $\delta$ is required to satisfy certain inequalities. But, with certainty, $\delta = 1$ can *never* be satisfied in the hypotheses of [19, Theorem 4].

Furthermore, positive Ricci curvature may have *downsides* when used for GNNs. One significant downside can be derived using the main result of [55], which says that the three properties of expansion, sparsity and non-negative Ollivier curvature are incompatible, in the following sense.

**Theorem 13.** *For any $\delta > 0$ and $\Delta > 0$, there are only finitely many graphs with maximum vertex degree $\Delta$, Cheeger constant at least $\delta$ and non-negative Ollivier curvature.*

We prove Theorem 13 in Appendix B. Furthermore, quoting directly from [55]:

*"The high-level message is that on large sparse graphs, non-negative curvature (in an even weak sense) induces extremely poor spectral expansion. This stands in stark contrast with the traditional idea – quantified by a broad variety of functional inequalities over the past decade – that non-negative curvature is associated with good mixing behavior."*

In our view, it is highly desirable that the graphs used for GNNs have high Cheeger constants, in the sense of globally lacking bottlenecks. Having bounded vertex degree is certainly useful too, since it implies that the graphs will be sparse, and the nodes will not have to handle ever-increasing neighbourhoods for message passing as graphs grow larger in size.

However, by proving Theorem 13, we showed non-negative Ollivier curvature is *incompatible* with these properties for sufficiently large graphs. Specifically, given the *finite* supply of non-negatively

curved sparse graphs, we can define $N'$ as the largest number of nodes of such graphs. Then, for all graphs $G$ where $|V(G)| > N'$, we will be *unable* to produce a computational graph for a GNN which is non-negatively curved everywhere. It remains an interesting challenge to provide an bound on $N'$ (as a function of $\delta$ and $\Delta$). It is possible that a careful analysis of [55] may provide this.

Further, while the expander graphs we generate are negatively-curved at $-1$ everywhere, and we will empirically show this helps alleviate oversquashing, we also believe that it is worthy of further investigation to theoretically examine whether performance of GNNs decreases significantly when the curvature is less than $-1$.

The negative curvature of each edge in $G_n$ implies that they are locally 'tree-like'. In Appendix C, we make this statement precise by showing that $G_n$ is 'tree-like' up to scale $c \log(n)$ about each node, for $c \simeq (1/2)(\log((1 + \sqrt{5})/2))^{-1}$ (see Figure 1 (Right) for a schematic view).

This tree-like structure might seem, at first, to be counter-productive for good propagation across the graphs $G_n$. Indeed, GNNs based on trees have been shown to have provably poor performance [18]. The reason for this seems to be two-fold. On the one hand, trees have small Cheeger constant. Indeed, any tree $G$ on $n$ vertices has a Cheeger constant $1/\lfloor n/2 \rfloor$, since we may find an edge that, when removed, decomposes the graph into subgraphs with $\lfloor n/2 \rfloor$ and $\lceil n/2 \rceil$ vertices. As discussed in Section 3 and in [19], when a graph has small Cheeger constant, its performance when used as a template for a GNN is likely to become poor. Secondly, GNNs based on trees are susceptible to oversquashing. For a $k$-regular infinite tree, there are $k(k-1)^{r-1}$ vertices at distance $r$ from a given vertex. Hence, if information is to be propagated at least distance $r$ from a given vertex, then seemingly an exponential amount of information is required to be stored.

However, neither of these issues are problematic for a GNN based on the Cayley graph $G_n$. By Theorem 9, their Cheeger constants are bounded away from 0. Secondly, although they are tree-like locally, this is only true up to scale $O(\log n)$. In fact, the $r$-neighbourhood of any vertex is the whole graph $G_n$ as soon as $r > C \log n$, for some constant $C$, by Theorem 5. Being tree-like up to distance $O(\log n)$ does not lead to a requirement to store too much information as the message propagates. This is because $k(k-1)^{r-1}$ is polynomial in $n$ when $r \leqslant O(\log n)$. Beyond this scale, there exist many additional connections, which lead to many possible paths joining any pair of vertices. The perspective of information transfer also gives rise to another perspective in which expanders fare very favourably: the *mixing time* of their corresponding Markov chain (see Appendix D for details).

## 5 Expander graph propagation

Let an input to a graph neural network be a node feature matrix $\mathbf{X} \in \mathbb{R}^{|V| \times d}$, and an adjacency matrix $\mathbf{A} \in \mathbb{R}^{|V| \times |V|}$. This setup is such that the feature vector of node $u$, $\mathbf{x}_u \in \mathbb{R}^d$, can be recovered by taking an appropriate row from $\mathbf{X}$. Note that the adjacency information can also be fed in an edge-list manner, which is desirable from a scalability perspective. Further, each edge in the graph may be endowed with additional features rather than a single real scalar. None of the above modifications would change the essence of our findings; we use a matrix formalism here purely for simplicity.

There exist many ways in which the computed Cayley graph $\mathrm{Cay}(\mathrm{SL}(2, \mathbb{Z}_n); S_n)$ can be leveraged for message propagation, and exploring these variations could be very useful for future work. Here, we opt for a simple construction: interleave running a standard GNN over the given input structure, followed by running another GNN layer over the relevant Cayley graph. If we let $\mathbf{A}^{\mathrm{Cay}(n)}$ be an adjacency matrix derived from $\mathrm{Cay}(\mathrm{SL}(2, \mathbb{Z}_n); S_n)$, this implies:

$$\mathbf{H} = \mathrm{GNN}(\mathrm{GNN}(\mathbf{X}, \mathbf{A}), \mathbf{A}^{\mathrm{Cay}(n)}) \tag{14}$$

Here, GNN refers to any preferred GNN layer, such as the graph isomorphism network [56, GIN]:

$$\mathbf{h}_u = \phi \left( (1 + \epsilon) \, \mathbf{x}_u + \sum_{v \in \mathcal{N}_u} \mathbf{x}_v \right) \tag{15}$$

where $\mathcal{N}_u$ is the neighbourhood of node $u$, i.e. in our setup, the set of all nodes $v$ such that $a_{vu} \neq 0$. $\epsilon \in \mathbb{R}$ is a learnable scalar, and $\phi \colon \mathbb{R}^d \to \mathbb{R}^{d'}$ is a two-layer MLP.

This procedure is iterated for a certain number of steps, after which the computed node embeddings in $\mathbf{H}$ can be used for any downstream task of interest—such as node classification, link prediction

or graph classification. Note that, unlike [18], who apply their custom layer only at the *tail* of the architecture, we apply the expander graph immediately after each layer over the input graph. We find that if the input graph given by $\mathbf{A}$ contains bottlenecks, applying the GNN over $\mathbf{A}^{\mathrm{Cay}(n)}$ only at the end may result in oversquashing occurring before any expander graph propagation can take place.

The setup so far assumed the number of nodes in our input graph to line up with the Cayley graph, that is, $\mathbf{A}^{\mathrm{Cay}(n)} \in \mathbb{R}^{|V| \times |V|}$. However, there is no guarantee that we can find an appropriate $n$ such that $\mathrm{Cay}(\mathrm{SL}(2, \mathbb{Z}_n); S_n)$ would have $|V|$ nodes. What we can do in practice, as an approximation, is choose the smallest $n$ such that the number of nodes of $\mathrm{Cay}(\mathrm{SL}(2, \mathbb{Z}_n); S_n)$ is $\geq |V|$, then consider $\mathbf{A}^{\mathrm{Cay}(n)}_{1:|V|,1:|V|}$—i.e. only the subgraph containing the first $|V|$ nodes in the Cayley graph.

There is a slight misalignment to our theory in this slicing choice—if the $|V|$ vertices in this subgraph are chosen completely arbitrarily, we risk disconnecting the graph. However, in all our experiments we construct the Cayley graph in a breadth-first manner, starting from the identity element as "node zero". Hence, the node at index $i$ is always guaranteed to be reachable from the nodes at lower indices ($j < i$), and the graph cannot be disconnected under this construction. More interesting strategies for this step can also be considered in the future. Note that, much like the fully connected graph used by [18], we interpret the Cayley graph mainly as a *template* for global information propagation, in order to relieve bottlenecks in a scalable way. Our interpretation, hence, assumes that the efficient diffusion of information over the whole graph is of benefit to the learning task we perform. When this is not the case, it might be worthwhile to construct expanders that somehow align with the input graph, but no such expander constructions are currently known, to the best of our knowledge. There is also a possible effect of *stochasticity* due to arbitrarily having to align the Cayley graph's nodes to the input graph—which would not appear when using master nodes or fully-connected graphs—though our preliminary experiments did not observe any such negative effects.

Algorithm 1 summarises the steps of our proposed EGP model. As direct corollaries of results we proved or demonstrated, we note that EGP satisfies all four of our desirable criteria: (**C1**) by Theorem 5 (so long as logarithmically many layers are applied), (**C2**) by Theorem 4 (high Cheeger constant implies no bottlenecks), (**C3**) by the fact our Cayley graphs are 4-regular and hence sparse, and (**C4**) by the fact we can generate a Cayley graph of appropriate size without detailed analysis of the input—we may precompute a "bank" of Cayley graphs of various sizes to use in an ad-hoc manner.

---

**Algorithm 1:** Expander graph propagation (EGP) forward pass

**Inputs :** Node features $\mathbf{X} \in \mathbb{R}^{|V| \times d}$, Adjacency matrix $\mathbf{A} \in \mathbb{R}^{|V| \times |V|}$
**Output :** Node embeddings $\mathbf{H}$

// Choose the smallest Cayley graph from our family that has number of nodes equal to, or greater than, $|V|$
$n \leftarrow \arg\min_{m \in \mathbb{N}} |V(\mathrm{Cay}(\mathrm{SL}(2, \mathbb{Z}_m); S_m))| \geq |V|$;   // We can use Equation 10 to determine $n$

$G^{\mathrm{Cay}(n)} \leftarrow \mathrm{Cay}(\mathrm{SL}(2, \mathbb{Z}_n); S_n)$

$\mathbf{A}^{\mathrm{Cay}(n)}_{uv} \leftarrow \begin{cases} 1 & (u,v) \in E(G^{\mathrm{Cay}(n)}) \\ 0 & \text{otherwise} \end{cases}$;   // Populate adjacency matrix of the Cayley graph

$\mathbf{H}^{(0)} \leftarrow \mathbf{X}$;               // Initialise GNN inputs

**for** $t \in \{1, \dots, T\}$ **do**
    **if** $t \mod 2 = 0$ **then**
        $\mathbf{H}^{(t)} \leftarrow \mathrm{GNN}^{(t)}(\mathbf{H}^{(t-1)}, \mathbf{A})$ ;   // GNN layer over input graph; e.g. Equation 15
    **end**
    **else**
        $\mathbf{H}^{(t)} \leftarrow \mathrm{GNN}^{\mathrm{Cay}(t)}\left(\mathbf{H}^{(t-1)}, \mathbf{A}^{\mathrm{Cay}(n)}_{1:|V|,1:|V|}\right)$;  // GNN layer over Cayley graph; e.g. Eq. 15
    **end**
**end**

**return** $\mathbf{H}^{(T)}$ ;         // Return final embeddings for downstream use

---

**Table 2:** Comparative evaluation performance on the four datasets studied. Our baseline model is a GIN [56], using exactly the same implementation as in [57]. See Appendix E for ablations.

| Model | ogbg-molhiv | ogbg-molpcba | ogbg-ppa | ogbg-code2 |
|---|---|---|---|---|
| GIN | $0.7558 \pm 0.0140$ | $0.2266 \pm 0.0028$ | $0.6892 \pm 0.0100$ | $0.1495 \pm 0.0023$ |
| GIN + EGP | $\mathbf{0.7934} \pm 0.0035$ | $\mathbf{0.2329} \pm 0.0019$ | $\mathbf{0.7027} \pm 0.0159$ | $\mathbf{0.1497} \pm 0.0015$ |

## 6 Empirical evaluation

Our work provides mainly a theoretical contribution: demonstrating a simple, theoretically-grounded approach to relieving bottlenecks and oversquashing in GNNs without requiring quadratic complexity or dedicated preprocessing. Further, we prove several additional results which deepen our understanding of curvature-based analysis of GNNs, showing how our expanders can be favourable in spite of their negatively-curved edges. We now provide results that empirically supplement our claim.

**Tree-NeighborsMatch**  We start by comparing our models on the `Tree-NeighborsMatch` task (for more details, see Alon and Yahav [18, Section 4.1]). `Tree-NeighborsMatch` is a synthetic benchmark explicitly designed to test a GNN's ability to counter oversquashing, and therefore it allows us to empirically verify that EGP is capable of alleviating oversquashing. We augment the original GIN implementation from the authors [18] with EGP layers, and find that it is capable of solving the task at `depth=5` **at 100% accuracy**, demonstrating alleviated oversquashing. In comparison, baseline GIN can only achieve 29% on this same task, and the best-performing GNN without EGP—i.e. propagating over the input tree only—cannot exceed 60% accuracy.

**OGB Datasets**  For real-world evaluation, we leverage the established Open Graph Benchmark collection of tasks [57, OGB]. Specifically, we provide results on all of its graph classification datasets: `ogbg-molhiv`, `ogbg-molpcba`, `ogbg-ppa` and `ogbg-code2`. The first two are among the largest molecule property prediction datasets in the MoleculeNet benchmark [58]. The third dataset is concerned with classifying species into their taxa, from their protein-protein association networks [59, 60] given as input. The fourth dataset is a *code summarisation* task: it requires predicting the tokens in the name of a Python method, given the abstract syntax tree (AST) of its implementation.

We provide a summary of important dataset statistics in Appendix E; please see [57] for detailed information. These datasets are designed to span a wide variety of domains (virtual drug screening, molecular activity prediction, protein-protein interactions, code summarisation) and sizes (from small molecules to very large syntax trees—the largest graph in `ogbg-code2` has $36,123$ nodes).

**Models**  In all four datasets, we want to *directly* evaluate the empirical gain of introducing an EGP layer and completely rule out any effects from parameter count, or similar architectural decisions.

To enable this, we take inspiration from the experimental setup of [18]. Our baseline model is the GIN [56], with hyperparameters as given by [57]. We use the *official* publicly available model implementation from the OGB authors [57], and modify all *even* layers of the architecture to operate over the appropriately-sampled Cayley graph.

Note that our construction leaves both the parameter count and latent dimension of the model *unchanged*, hence any benefits coming from optimising those have been diminished.

**Results**  The results of our evaluation are presented in Table 2. It can be observed that, in all four cases, propagating information over the Cayley graph yields improvements in mean performance—these improvements are most apparent on `ogbg-molhiv`, but also present in `ogbg-molpcba` and `ogbg-ppa`. We believe that these results provide encouraging empirical evidence that propagating information over Cayley graphs is an elegant idea for alleviating bottlenecks. We provide additional results on OGB, comparing EGP to various other oversquashing-countering methods, in Appendix E.

## 7 Conclusion

In this paper, we have presented expander graph propagation (EGP), a novel and elegant approach to alleviating bottlenecks in graph representation learning, which provably supports global communication while not requiring quadratic complexity or dedicated preprocessing of the input.

To this end, we offered a detailed theoretical overview of Cayley graphs of special linear groups, $\mathrm{Cay}(\mathrm{SL}(2, \mathbb{Z}_n); S_n)$. We cite proofs that these graphs have highly favourable properties for information propagation in graph neural networks: they are sparse and $4$-regular, they have logarithmic diameter, and they can be efficiently precomputed by a simple procedure that does not rely on the input structure. We show that, in spite of having negatively curved edges, our findings do not violate any prior results on understanding oversquashing via curvature. Even under a simple intervention—interleaving EGP layers inbetween standard GNN layers—we have been able to recover significant performance returns without changing the parameter count or latent space dimensionality.

We hope that our work serves as a foundation for further work on deploying Cayley graphs—or other expander families—within the context of GNNs.

## Acknowledgements

We would like to thank the developers of the Open Graph Benchmark [57], whose open-source implementations made our experiments much easier to prepare. In addition, we are very grateful to Francesco Di Giovanni for his very thoughtful discussions about our work and its broader implications, and especially for bringing to our attention the important work from Salez [55]. We also thank Alex Davies, Bogdan Georgiev and Charles Blundell for reviewing the paper prior to submission, and all of our anonymous reviewers for their diligent feedback.

This research was conceived while all authors were visiting the Institute for Advanced Study (IAS) in Princeton. The surreal atmosphere at the IAS strongly stimulated the creative development of research ideas at the intersection of machine learning and pure mathematics. We would like to thank the IAS for kindly inviting us to visit, and hope to come again soon!

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

# A   Proof of Proposition 11

*Let $s$ be one of*

$$\begin{pmatrix} 1 & 1 \\ 0 & 1 \end{pmatrix}, \qquad \begin{pmatrix} 1 & 0 \\ 1 & 1 \end{pmatrix}.$$

*Let $n, n' > 18$ and let $e$ and $e'$ be $s$-labelled edges in $G_n$ and $G_{n'}$. Then there is a graph isomorphism between $N_2(e)$ and $N_2(e')$ taking $e$ to $e'$.*

*Proof.* Note first that, by the homogeneity of the Cayley graphs $G_n$ and $G_{n'}$, we may assume that $e$ and $e'$ emanate from the identity vertex of each graph.

Let $G_\infty$ be the Cayley graph of $\mathrm{SL}(2, \mathbb{Z})$ with respect to the generators

$$S_\infty = \left\{ \begin{pmatrix} 1 & 1 \\ 0 & 1 \end{pmatrix}, \begin{pmatrix} 1 & 0 \\ 1 & 1 \end{pmatrix} \right\}.$$

Let $e_\infty$ be the $s$-labelled edge emanating from the identity vertex of $G_\infty$. The quotient homomorphism

$$\mathrm{SL}(2, \mathbb{Z}) \to \mathrm{SL}(2, \mathbb{Z}_n)$$

induces a graph homomorphism $G_\infty \to G_n$ sending $e_\infty$ to $e$. We will show that it restricts to a graph isomorphism

$$N_2(e_\infty) \to N_2(e).$$

As there is a similar graph isomorphism $N_2(e_\infty) \to N_2(e')$, the proposition will follow.

Note that two elements of $\mathrm{SL}(2, \mathbb{Z})$ map to the same element of $\mathrm{SL}(2, \mathbb{Z}_n)$ if and only if they differ by multiplication by an element of the kernel $K_n$. This is

$$K_n = \left\{ \begin{pmatrix} a & b \\ c & d \end{pmatrix} \in \mathrm{SL}(2, \mathbb{Z}) : a \equiv d \equiv 1 \bmod n \text{ and } b \equiv c \equiv 0 \bmod n \right\}.$$

The graph homomorphism sends edges to edges, and so it is distance non-increasing. Hence it certainly sends $N_2(e_\infty)$ to $N_2(e)$. It is also clearly surjective, because any element of $N_2(e)$ is reached from an endpoint of $e$ by a path of length at most 2, and there is a corresponding path in $N_2(e_\infty)$.

We just need to show that this is an injection. If not, then two distinct vertices $g_1$ and $g_2$ in $N_2(e_\infty)$ map to the same vertex in $N_2(e)$. Note then that as elements of $\mathrm{SL}(2, \mathbb{Z})$, $g_2 = g_1 k$ for some $k \in K_n$. There are paths with length at most 3 joining the identity 1 to $g_1$ and $g_2$ respectively. Hence, the distance in $G_\infty$ between $g_1$ and $g_2$ is at most 6. Therefore, the distance between 1 and $g_1^{-1} g_2$ is at most 6. This element $g_1^{-1} g_2$ lies in $K_n$. We will show that when $n > 18$, the only element of $K_n$ that has distance at most 6 from the identity is the identity itself. This will imply that $g_1^{-1} g_2 = 1$ and hence $g_1 = g_2$. But this contradicts the assumption that $g_1$ and $g_2$ are distinct vertices. Our argument follows that of [61].

The operator norm $||A||$ of a matrix $A \in \mathrm{SL}(2, \mathbb{Z})$ is

$$||A|| = \sup\{|A(v)| : v \in \mathbb{R}^2, \ |v| = 1\}.$$

This is submultiplicative: $||AB|| \leqslant ||A|| \, ||B||$ for matrices $A$ and $B$. It can be calculated as the square root of the largest eigenvalue of $A^t A$. In our case, the operator norms satisfy

$$\left\| \begin{pmatrix} 1 & 1 \\ 0 & 1 \end{pmatrix} \right\| = \left\| \begin{pmatrix} 1 & 0 \\ 1 & 1 \end{pmatrix} \right\| = \frac{1 + \sqrt{5}}{2}.$$

Consider an element

$$K = \begin{pmatrix} a & b \\ c & d \end{pmatrix}$$

of $K_n$ that is not the identity. Since $a \equiv d \equiv 1$ modulo $n$ and $b \equiv c \equiv 0$ modulo $n$, we deduce that at least one $|a|$, $|b|$, $|c|$ and $|d|$ is at least $n - 1$. Therefore, this matrix acts on one of the vectors $(1, 0)^t$ or $(0, 1)^t$ by scaling its length by at least $n - 1$. Therefore, $||K|| \geqslant n - 1$. Suppose now that $K$ has distance at most 6 from the identity. Then $K$ can be written as a word in the generators of $\mathrm{SL}(2, \mathbb{Z})$ with length at most 6. Therefore, we obtain the inequality

$$||K|| \leqslant \left( \frac{1 + \sqrt{5}}{2} \right)^6 < 17.95.$$

Hence, $n < 18.95$ and therefore, as $n$ is integral, $n \leqslant 18$. $\qquad \square$

## B  Proof of Theorem 13

*For any $\delta > 0$ and $\Delta > 0$, there are only finitely many graphs with maximum vertex degree $\Delta$, Cheeger constant at least $\delta$ and non-negative Ollivier curvature.*

*Proof.* This is a consequence of the main result of Salez [55, Theorem 3]. This states if $G_n = (V_n, E_n)$ is a sequence of graphs with the following properties:

$$\sup_{n \geqslant 1} \left\{ \frac{1}{|V_n|} \sum_{v \in V_n} \deg(v) \log \deg(v) \right\} < \infty \tag{16}$$

$$\forall \epsilon > 0, \quad \frac{1}{|E_n|} |\{e \in E_n : \kappa(e) < -\epsilon\}| \to 0 \text{ as } n \to \infty, \tag{17}$$

then

$$\forall \rho < 1, \quad \liminf_{n \to \infty} \left\{ \frac{1}{|V_n|} |\{i : \mu_i(G_n) \geqslant \rho\}| \right\} > 0.$$

Here, $\kappa(e)$ is the Ollivier curvature of an edge $e$ and

$$1 = \mu_0(G) \geqslant \mu_1(G) \geqslant \cdots \geqslant 0$$

are the eigenvalues of the lazy random walk operator. To prove the theorem, we suppose that on the contrary, there are infinitely many distinct graphs $G_n = (V_n, E_n)$ with maximum vertex degree $\Delta$, Cheeger constant at least $\delta$ and non-negative Olliver curvature. Then

$$\sum_{v \in V_n} \deg(v) \log \deg(v) \leqslant |V_n| \Delta \log \Delta$$

and so condition 16 is satsfied. Condition 17 is trivially satisfied because the Ollivier curvature of each graph is non-negative. Thus, we deduce that the conclusion of Salez' theorem holds. Setting $\rho = 1 - (\delta^2/4\Delta^2)$, we deduce that a definite proportion of the eigenvalues of the lazy random walk operator are at least $1 - (\delta^2/4\Delta^2)$. In particular, $\mu_1(G_n) \geqslant 1 - (\delta^2/4\Delta^2)$. Denote the eigenvalues of the normalised Laplacian by

$$0 = \lambda_0'(G_n) \leqslant \lambda_1'(G_n) \leqslant \ldots$$

These are related to the eigenvalues of the lazy random walk operator by $\lambda_i'(G_n) = 2 - 2\mu_i(G_n)$. Hence, $\lambda_1'(G_n) \leqslant \delta^2/(2\Delta^2)$. There is a variation of Cheeger's inequality that relates $\lambda_1'$ to the *conductance* of the graph. To define this, one considers subsets $A$ of the vertex set, and defines their *volume* to be $\mathrm{vol}(A) = \sum_{v \in A} \deg(v)$. The conductance $\phi(G)$ of a graph $G$ is

$$\phi(G) = \min \left\{ \frac{|\partial A|}{\mathrm{vol}(A)} : A \subset V(G), \, 0 < \mathrm{vol}(A) \leqslant \mathrm{vol}(V(G))/2 \right\}.$$

Then, by Chung [42, Theorem 2.2],

$$\phi(G) \leqslant \sqrt{2\lambda_1'(G)}$$

Hence, in our case,

$$\phi(G_n) \leqslant \delta/\Delta.$$

Consider any subset $A_n$ of the vertex set that realises $\phi(G_n)$. Thus $0 < \mathrm{vol}(A_n) \leqslant \mathrm{vol}(V_n)/2$ and $|\partial A_n|/\mathrm{vol}(A_n) = \phi(G_n) \leqslant \delta/\Delta$. If $A_n$ is at most half the vertices of $G_n$, then this implies that the Cheeger constant $h(G_n) \leqslant \delta$. On the other hand, if $A_n$ is more than half the vertices of $G_n$, we consider its complement $A_n^c$. Its cardinality $|A_n^c|$ satisfies

$$|A_n^c| \geqslant \mathrm{vol}(A_n^c)/\Delta.$$

Hence,

$$h(G_n) \leqslant \frac{|\partial A_n^c|}{|A_n^c|} \leqslant \frac{|\partial A_n|\Delta}{\mathrm{vol}(A_n^c)} \leqslant \frac{|\partial A_n|\Delta}{\mathrm{vol}(A_n)} = \phi(G_n)\Delta \leqslant \delta.$$

In either case, we deduce that the Cheeger constant of $G_n$ is at most $\delta$, contradicting one of our hypotheses. Hence, there must have been only finitely many graphs satisfying the conditions of the theorem. $\qquad \square$

## C   Cayley graph at infinity is quasi-isometric to a tree

As all vertices of $G_n$ look the same, we focus attention on $N_r(1)$, the $r$-neighbourhood of the identity vertex. The proof of Proposition 11 immediately gives the following.

**Proposition 18.** *Let $r$ be a positive integer satisfying*

$$r < \frac{1}{2} \left( \log \left( \frac{1 + \sqrt{5}}{2} \right) \right)^{-1} \log(n - 1).$$

*Then there is a graph isomorphism between the $r$-neighbourhood of the identity vertex in $G_n$ and the $r$-neighbourhood of the identity vertex in $G_\infty$. This isomorphism takes the identity vertex to the identity vertex.*

*Proof.* As shown in the proof of Proposition 11, there is a graph homomorphsm from $N_r(1)$ in $G_\infty$ to $N_r(1)$ in $G_n$ that is a surjection. If it fails to be an injection, then there is a non-trivial element $K$ in the kernel $K_n$ of $\mathrm{SL}(2, \mathbb{Z}) \to \mathrm{SL}(2, \mathbb{Z}_n)$ satisfying

$$||K|| \leqslant \left( \frac{1 + \sqrt{5}}{2} \right)^{2r}.$$

But any non-trivial element $K$ in $K_n$ satisfies

$$||K|| \geqslant n - 1.$$

Rearranging gives the required inequality. $\qquad\qquad\square$

This raises the question of the local structure of $G_\infty$. The answer is well-known: it is 'tree-like'. Specifically, it is quasi-isometric to a tree. The formal definition of quasi-isometry is as follows.

**Definition 19.** A *quasi-isometry* between two metric spaces $(X_1, d_1)$ and $(X_2, d_2)$ is a function $f \colon X_1 \to X_2$ that satisfies the following two conditions:

1. there are constants $c, C > 0$ such that, for every $x, x' \in X_1$

$$c \, d_1(x, x') - c \leqslant d_2(f(x), f(x')) \leqslant C \, d_1(x, x') + C,$$

2. there is a constant $K \geqslant 0$ such that for every $y \in X_2$, there is an $x \in X_1$ with $d_2(f(x), y) \leqslant K$.

If there is such a quasi-isometry, we say that $(X_1, d_1)$ and $(X_2, d_2)$ are *quasi-isometric*.

This forms an equivalence relation on metric spaces. When two metric spaces are quasi-isometric, they are viewed as being 'essentially the same' at large scales.

When $S$ and $S'$ are finite generating sets for a group $\Gamma$, the graphs $\mathrm{Cay}(\Gamma; S)$ and $\mathrm{Cay}(\Gamma; S')$ are quasi-isometric. Hence, the quasi-isometry type of a finitely generated group is well-defined, and this is the central object of study in geometric group theory.

The group $\mathrm{SL}(2, \mathbb{Z})$ has a finite-index subgroup that is a free group $F$ [62]. If $S'$ denotes a free generating set for $F$, then $\mathrm{Cay}(F; S')$ is a tree. As passing to a finite-index subgroup preserves its quasi-isometry class, we deduce that the Cayley graph $G_\infty = \mathrm{Cay}(\mathrm{SL}(2, \mathbb{Z}); S_\infty))$ is indeed quasi-isometric to a tree, as claimed above.

## D   Mixing time properties of expander graphs

Expanders are well known to have small mixing time, in the following sense.

Let $G$ be a graph. We will consider probability distributions $\pi$ on $V(G)$. The lazy random walk operator $M$ acts on probability distributions as follows. We think of $\pi(v)$ as being the probability of the random walk being at vertex $v$. If the current location of the walk is at $v$, then at the next step of the walk, either we stay put with probability $1/2$ or we move to one of its neighbours with equal probability. Then $M\pi$ is the new probability distribution.

In the case when $G$ is $k$-regular, this takes a particular simple form. The operator $M$ is represented by the matrix $(1/2)I + (1/2k)A$, where $A$ is the adjacency matrix. In that case, any initial distribution $\pi$ converges under powers of $M$ to the uniform distribution.

This is true for any reasonable notion of convergence, but we will use the $\| \cdot \|_1$ norm, where for two probability distributions $\pi$ and $\pi'$,

$$\left\| \pi - \pi' \right\|_1 = \sum_{v \in V(G)} |\pi(v) - \pi'(v)|.$$

**Definition 20.** The *mixing time* for a regular graph $G$ is the minimum value of $\ell$ such that for any starting probability distribution $\pi$ on the vertex set of $G$,

$$\left\| M^\ell \pi - u \right\|_1 \leqslant \frac{1}{4}.$$

Here, $u$ is the uniform probability distribution on the vertex set, and $M$ is the lazy random walk operator.

**Table 3:** Statistics of the three graph classification datasets studied in our evaluation.

| Name | Number of graphs | Avg. nodes/graph | Avg. edges/graph | Metric |
|------|------------------|------------------|------------------|--------|
| ogbg-molhiv | $41,127$ | 25.5 | 27.5 | ROC-AUC |
| ogbg-molpcba | $437,929$ | 26.0 | 28.1 | Avg. precision |
| ogbg-ppa | $158,100$ | 243.4 | $2,266.1$ | Accuracy |
| ogbg-code2 | $452,741$ | 125.2 | 124.2 | $F_1$ score |

Expanders have small mixing times in the following very strong sense.

**Theorem 21.** *For any $k > 0$ and $\delta > 0$, there is a constant $c > 0$ with the following property. If $G$ is a connected $k$-regular graph on $n$ vertices with Cheeger constant at least $\delta > 0$, then the mixing time for $G$ is at most $c \log(n)$.*

# E  Additional experimental details and ablations

**OGB dataset statistics.**  We provide additional details on the dataset statistics for the OGB tasks we used in Table 3. More substantial details can be found in the OGB paper [57].

**Ablations on propagation graph.**  Our work concerns sparse expander graphs, determined using the Cayley graphs of the special linear group. We acknowledge that this approach, while theoretically beneficial, is not the only possible way to aid global information propagation in a GNN. Therefore, in this subsection we compare against other classes of approaches.

Our additional baseline methods include: GINs with a *master node*, GINs with a *fully connected* layer (FA), as done in Alon and Yahav [18], and GINs with applying a recently proposed rewiring method, G-RLEF [33].

Note that both the FA method and G-RLEF have motivations related to expanders: the fully-connected graph in the FA method is a trivial *dense* expander, whereas G-RLEF's rewiring iterations can converge to an expander for certain input graph distributions. Therefore, comparing against these methods allows us to also evaluate the impacts of expander density, as well as proximity to the input graph (since G-RLEF iteratively modifies the input graph). We run G-RLEF for $O(V)$ steps.

The results of our ablative analysis are summarised in Table 4. We find that, as expected, all of our added methods outperform the baseline GIN, demonstrating that oversquashing had been alleviated. When comparing them against each other, however, we find that EGP tends to be highly competitive on two out of the three datasets considered (having the largest average overall). The fully-adjacent dense expander method remains strong on both `ogbg-molhiv` and `ogbg-molpcba`, but runs out of memory as graphs increase in size (as is the case with `ogbg-ppa`).

We find that this collection of ablation studies further supplements the analysis of EGP we have conducted, and serves as a good starting point for further investigations of expander propagation templates with various properties.

**Table 4:** Comparative ablation performance of various propagation templates on `ogbg-molhiv`, `ogbg-molpcba` and `ogbg-ppa`. Our baseline model is a GIN [56], using exactly the same implementation as in [57]. All models have *exactly* the same number of parameters—we only modify the connectivity in certain layers depending on the scheme. **N.B.** The fully-connected graph, used in the FA approach [18] can be seen as a dense expander graph, i.e. a special case of EGP. 'OOT' indicates that the method failed to approach baseline performance within five days of training time (while not converging within this time), and 'OOM' indicates out-of-memory (on a V100 GPU).

| Model | ogbg-molhiv | ogbg-molpcba | ogbg-ppa |
|---|---|---|---|
| GIN | $0.7558 \pm 0.0140$ | $0.2266 \pm 0.0028$ | $0.6892 \pm 0.0100$ |
| GIN + master node | $0.7668 \pm 0.0096$ | $0.2527 \pm 0.0064$ | $0.6916 \pm 0.0154$ |
| GIN + FA [18] | $0.7850 \pm 0.0090$ | $\mathbf{0.2595} \pm 0.0049$ | OOM |
| GIN + G-RLEF [33] | $0.7802 \pm 0.0024$ | OOT | OOM |
| GIN + EGP (ours) | $\mathbf{0.7934} \pm 0.0035$ | $0.2329 \pm 0.0019$ | $\mathbf{0.7027} \pm 0.0159$ |

