# OpenReview forum: "Expander Graph Propagation"
_logconference.io/LOG/2022/Conference — LoG 2022 Poster_

### Official Review · Reviewer_yhzE · 2022-09-24

**Overall Score:** 6
**Confidence:** 4

**Review:**

The paper proposed a way to use an expander graph interleaved with the original graph to facilitate message passing GNN. Compared to other related work such as stacking more layers, adding a master node, graph transformer, graph rewiring, and hierarchical message passing, the expander graph is a quite simple and straight way and satisfies all nice properties (C1-C4).

## Strong points:
simple approach. Elegant idea. Nice summary of related work.
Good background on expander graph. I consider it a nice contribution to introducing expander graphs, a very important topic in theoretical computer science, into the graph learning community.

## Weak points:
- My main concern is that the empirical study is a bit thin. I am quite interested in how the expander graph approach is compared to other approaches such as virtual nodes or graph transformers. I suspect they also bring in some advantages so I’d like to see more experiments on comparing different approaches.

- https://openreview.net/forum?id=in7XC5RcjEn&referrer=%5Bthe%20profile%20of%20Dominique%20Beaini%5D(%2Fprofile%3Fid%3D~Dominique_Beaini1) I’d like to draw attention to the recent dataset paper. They proposed some datasets where modeling long-range interaction is crucial for better performance. I expect on those datasets proposed method may demonstrate a larger improvement.


I encourage authors can opensource the code. Expander graph construction is a handy feature that can be incorporated in dgl or pyg library.

---

### Official Review · Reviewer_Vpo6 · 2022-10-20

**Overall Score:** 3
**Confidence:** 4

**Review:**

This paper suggests augmenting graph neural networks with (sparse) expander graphs in order to overcome bottlenecks and oversquashing during information propagation. Specifically, the expanders used in these papers are based on Cayley graphs of the special linear matrix group, $SL(2, \mathbb{Z}_n)$. The authors overlay the original graph $G$ with a Cayley graph $G’$ as above of (roughly) the same size, where the structure of $G’$ is independent of that of $G$. The GNN itself simply interleaves layers that use the input graph with layers of the expander graph. The authors run a limited empirical evaluation showing that their method slightly outperforms vanilla GNNs on graphs classification tasks.

The authors also discuss the interaction between expansion and (negative) curvature in the context of GNNs, showing in particular that the Ricci and Ollivier curvatures of the Cayley graph they construct are -1 and -0.5 for when $G$ is large enough. This is to be expected and in some sense desirable: While existing work shows that a Ricci curvature around $-2$ is problematic for propagation, it is also known that large graphs cannot simultaneously have sparsity (bounded maximum degree), good expansion (large Cheeger constant), and Olliver curvature close to zero.

Evaulation: Using expander graphs to overcome bottlenecks is a natural and attractive idea, but it is not new (see the papers pointed out by the AC, which seem very relevant; I wasn’t able to find other papers on this topic). While this paper is well-written and provides a good account of related literature, I believe that it lacks novelty and significant theoretical contributions as a standalone contribution. First, the construction of expanders based on Cayley graphs is a very standard practice (see, e.g., [HLW], Section 11). The discussion connecting curvature and expansion is interesting, but it is not well-developed enough. Many questions remain unanswered:
* The discussion on “good” curvature values remains at a conceptual level and is missing concrete theoretical upper and/or lower bounds that would help us understand it better. How should we interpret the curvature values that you obtained your computations? would it be better if they were closer to zero? are we “safe” from results such as those of [19] (possibly obtained through other techniques), or are there indications that Ricci curvature of 1 can be problematic?
* You computed the curvature of the expander graph, but the actual GNN combines expander and “original” layers. How does this combination affect information propagation?
* Additional empirical results are needed to better understand the effects of expansion in the results of Tables 2,3. The existing experimental results do not provide much insight, and it would be much better if, for example, you compared different types of expanders, including possibly expanders whose structure depends on that of the original graph.
* To continue the previous point: although the fact that the expander is completely independent of the structure of the graph is attractive in terms of implementation, I conjecture that this kind of choice is suboptimal in terms of efficacy. It would be good to look into this conjecture.
I believe that providing better answers/explanations to one or more of the above points will substantially improve the paper.

Comments:
- Section 3 (as a whole) is very standard and well known. Perhaps you should only cite Theorem 9 and move the rest of it to appendix.
- Theorem 13: did you mean “any small enough $\varepsilon > 0$”? otherwise this contradicts Proposition 12.
- End of Section 4, line 238: the discussion surrounding “does not lead to a requirement to store too much information…” — constants in the $O(\log n)$ matter here, since they go into the exponent. Try to make your statements more careful.
- Abstract, line 9: “we provide an efficient method…” — this method to construct expander graphs is well known (as you also hint in the main body of the paper, so the statement that you provide a method is misleading.

[HLW] S. Hoory, N. Linial, A. Wigderson, “Expander graphs and their applications”, Bulletion of the American Mathematical Society 43(4), 2006, pages 439-561.

---

### Official Review · Reviewer_kzW5 · 2022-10-22

**Overall Score:** 3
**Confidence:** 4

**Review:**

# Summary

This paper proposes to solve the over-squashing problem in graph neural networks (GNNs) by adding a layer of information propagation on the Cayley graph after every normal GNN layer. The major motivation is that the information propagation over the expander graphs has faster mixing time so it helps obtain global information within a fixed number of propagation layers. Theoretical analyses on the expander family and negative curvature are presented, followed by some empirical experiments.

# Strength

1. The proposed model is a novel and interesting way to potentially solve the over-squashing problem in GNNs.
2. The proposed EGP is efficient and only requires subquadratic complexity.
3. The findings and discussions from the perspective of negative curvature are interesting and insightful.
4. It is empirically verified that the proposed EGP can improve GIN on some datasets on graph classification problems.

# Weakness

1. The paper makes overclaims without substantial evidence. For instance, it is claimed that expander graphs have higher Cheeger constants and will hence provably be bottleneck-free (line 142). However, there is a lack of discussion about the exact meaning (or measurement) of the bottleneck. The claim is quite vague without such a clear definition and a quantitative measure (i.e., how large the Cheeger constant $\epsilon$ should be to make it bottleneck-free (Theorem 4). In other words, if a small Cheeger constant indicates a bottleneck, how small is it sufficient to cause the bottleneck?

2. The proposed idea is simple and efficient, i.e., adding a propagation layer on the expander graph. Although it helps propagate information faster over the whole graph which intuitively provides more global information in a larger receptive field, the negative impact of this propagation is not discussed or evaluated. How can EGP make sure that the structure information from the expander graph does not dominate and destroy the structure information in the original graph? In fact, this might cause other types of bottlenecks (such as over smoothing). More discussion on the negative aspect will be helpful.

3. The empirical evaluation of the proposed model is quite limited. It only shows that the GIN + EGP outperforms GIN, but how about other GNN models or other tasks such as link prediction or node classification? Furthermore, there is no deep study on whether the proposed EGP really solves the over-squashing issue. For instance, paper [1] provides multiple examples to demonstrate such a bottleneck. These kinds of experiments and analyses are necessary to demonstrate whether EPG really solves the over-squashing and becomes bottleneck-free.

4. The paper does not present a sufficient and comprehensive comparison with existing works that also mitigate the over-squashing problem.

5. There is a lack of discussion on the impact of the density of expander graphs (such as with more edges).

[1] On the Bottleneck of Graph Neural Networks and its Practical Implications

---

### Official Review · Reviewer_drzX · 2022-10-22

**Overall Score:** 6
**Confidence:** 4

**Review:**

In the submitted manuscript, the authors propose to use an expander graph construction to efficiently propagate information over the nodes of an input graph. This gives rise to a GNN model which alternates between message passing over the original graph structure and the constructed expander graph structure. The authors demonstrate how this altered message passing scheme addresses the issue of oversquashing and bottleneck edges. The proposed architecture leads to improvements in classification accuracy on four graph classification datasets in the OGB collection.

I would like to congratulate the authors on this very nice manuscript. It is insightful, well-written and clear (if the reader is sufficiently familiar with the background material). Your result on negatively curved edges is likely to be of large interest to the Graph Representation Learning community. I think your paper could benefit from extensions to your discussion of expander graph theory and several theoretical results could be strengthened. I furthermore believe that a discussion of the limitations of the proposed propagation scheme would be beneficial. I therefore, recommend a "weak accept" with the intention to raise my score if my comments are resolved. I list my detailed comments below.


1) I think your paper would be more accessible to readers if you briefly defined the problems of "oversquashing" and "bottlenecks" early on in the paper. Especially the second of these has a non-distinct name, which might confuse some readers not overly familiar with the graph representation learning literature.

2) In Definition 2 you define a collection of graphs {$G_i$} to be an expander family if there exists a $c$ such that for all $G_i$ $\lambda_1(G_i)>c.$ However, it is a well-known fact that for any connected graph $\lambda_1(G_i)>0$ (e.g., Proposition 2 in [1]). Therefore, you define any collection of connected graphs to be an expander family. I suggest talking about "$c$-expander families" to explicitly measure the quality of expanders in the name or resorting to an alternative definition of expander graphs from the literature [2,3].

3) It also seems to be the case that the statement you make in Theorem 4 is trivial. If the graph is connected then every considered vertex set has a boundary set with cardinality at least 1. This implies that the Cheeger constant of any connected graph is positive, which is your claim in Theorem 4. The statement you want to make here is true. But the Theorem is not sufficiently strong to evidence your claim. I think you might be able to make good use of the Cheeger Inequality [4, p.186] or [5, p.95], which lower bounds the expansion ratio by the second smallest Laplacian eigenvalue.

4) On Page 5 Line 147 you state that you have shown that expander graphs are "bottleneck-free". I think this may be worded too strongly to be true. Since, the concept of bottlenecks was not quantiatively defined by you, its absence can clearly be debated. I think it may be more accurate to say that the larger the second smallest Laplacian eigenvalue and Cheeger constant the less severe are problems arising due to bottleneck edges.

5) I am unsure how to interpret Theorem 13. What is the significance of only having finitely many graphs with the properties you lay out? And how does it show an incompatibility (Line 206) of the requirements?

6) If I understand correctly, then in your proposed GNN^{Cay} you propagate information between nodes in the graph without any consideration of the original graph structure. You lay out clearly how your expander graphs lead to an efficient propagation scheme. However, I think you should also mention that your model is based on the assumption that the efficient diffusion of information over the whole graph is of benefit to the learning task we perform. In situations where community structure in the graph aligns with the labels we are trying to predict, I could imagine your propagation scheme to be detrimental, because we are perfectly happy to not have information flow between graph communities. I think this limitation of your model might be worth discussing in your paper and it may be interesting to explore how the expander graph can be aligned with the original graph structure to limit this drawback.

7) Considering more GNNs in your experiments in Table 3 would help establish the practical relevance of your model beyond the GIN. To me this seems worthwhile.


8) Minor Comments:

8.1) Typo: Page 2 Line 59 "prior art"

8.2) On Page 1 Line 31 you write "Under many industrially-relevant tasks ([...]), this formalism is very well aligned, [...]". It unclear to me what discussed formalism aligns with. I suggest making minor edits to clarify this.

8.3) In Definition 1 $f$ is defined to map to $\mathbb{R}$ then $Lf$ is defined to map to $\mathbb{R}$ as well, which implies that $L$ maps from $\mathbb{R}$ to $\mathbb{R}.$ This is counter-intuitive and you rightfully state immediately following Definition 1 that $L$ maps from  $\mathbb{R}^n$ to $\mathbb{R}^n,$ where $n$ is the number of vertices. I suggest defining $Lf$ to map to $\mathbb{R}^n$ and to then only consider one vector entry in the display equation contained in Definition 1.

8.4) On several occasions you use the vertex set instead of its cardinality, e.g., Page 4 Line 125-6 "$\mathbb{R}^{V(G)}$" instead of "$\mathbb{R}^{|V(G)|}$" and Page 6 Footnote 2. You furthermore denote the vertex set by both "$V(G)$" and "$V$", using a single notation might be nicer.

8.5) On Page 6 Line 192 the notation $G_n$ is not clearly defined and in fact slightly confusing in the context of the notation $G_i$ used in Line 129, where the domain of i is also not defined.

8.6) On Page 7 Line 231, Line 239 and Line 247 you use $k$ to both denote the degree of a regular tree and the dimension of the node features. It might nice to resolve this clash.

8.7) The superscirpts $GNN^{(t)}$ and $GNN^{Cay}$ could be misunderstood to indicate that the trainable weights of all expander propagation layers are shared, since only the superscript of the standard GNN layer depends on $t.$ It might be nice to make a minor edit here.

---

### Meta-Review · Area_Chair_GMor · 2022-11-08

**Confidence:** 3
**Recommendation:** Accept

**Meta Review:**

The paper proposes alternating between message passing on the original input graph and *expander graphs*, to facilitate message passing and avoid oversquashing, with the motivation that the expander graph layers can be used as "shortcuts" for long-range information propagation.
The paper also argues that, in contrast with the common belief, negative curvature may not actually be problematic in principle, and even be unavoidable.

The reviewers have raised the following main weaknesses:
1. **Empirical evaluation: The approach was only demonstrated on GIN. Additionally, the authors could evaluate their approach on long-range datasets such as the "Long Range Graph Benchmark"** -- During the discussion period, the authors improved the paper by comparing with additional oversquashing-preventing approaches, and evaluating the approach on the synthetic Tree-NeighboursMatch task. This has strengthened the empirical evaluation, although the few datasets and evaluating only on GIN remain a weakness.
2. **The construction of expander graphs is a standard practice** -- Although the construction of expander graphs is not new, I believe that their application to GNNs will be useful and valuable to the community.
3. **The theoretical part regarding Ricci curvature values is worth further investigation** - I agree with the reviewers that the theoretical findings are worth further investigation, but I believe that they might be sufficient, and may spark future directions, which the authors added explicitly.
4. **The paper's close relatedness to Banerjee et al. (2022)** -- I consider the Banerjee et al. (2022) paper to be concurrent work, and therefore I do not weigh this as a significant weakness. Further, the authors have performed an empirical comparison that showed the advantages over Banerjee et al. in both accuracy and efficiency (in terms of memory and time).
5. **No comparison to graph pooling methods** -- this comment was brought up one day before the end of the revision period.
The authors performed an empirical comparison to a "master node" which is a simple pooling method,
the authors responded the next day with an explanation regarding different pooling methods and their disadvantages,
and they pointed to a discussion regarding pooling methods in the Related Work section, that existed since their original submission.
Thus, I did not count this argument against the paper, but I encourage the authors to include such additional baselines.

Overall, there could be an argument made that the paper would be stronger in a re-submission with improved empirical evaluation. However, I believe that the oversquashing problem and the new insights regarding negatively curved edges are of high interest to the community, and I think that the paper can be accepted.

Finally, I wish to thank the reviewers for their excellent work in writing thorough reviews, paying attention to details, being very responsive, and actively engaging in discussion. I believe that the reviewers' suggestions have helped significantly improve the initial version.

---

### Decision · Program_Chairs · 2022-11-23

Accept (Poster)